# MULTI-ADVISOR REINFORCEMENT LEARNING

## ABSTRACT

We consider tackling a single-agent RL problem by distributing it to $n$ learners. These learners, called advisors, endeavour to solve the problem from a different focus. Their advice, taking the form of action values, is then communicated to an aggregator, which is in control of the system. We show that the local planning method for the advisors is critical and that none of the ones found in the literature is flawless: the *egocentric* planning overestimates values of states where the other advisors disagree, and the *agnostic* planning is inefficient around danger zones. We introduce a novel approach called *empathic* and discuss its theoretical aspects. We empirically examine and validate our theoretical findings on a fruit collection task.

## 1 INTRODUCTION

When a person faces a complex and important problem, his individual problem solving abilities might not suffice. He has to actively seek for advice around him: he might consult his relatives, browse different sources on the internet, and/or hire one or several people that are specialised in some aspects of the problem. He then aggregates the technical, ethical and emotional advice in order to build an informed plan and to hopefully make the best possible decision. A large number of papers tackle the decomposition of a single Reinforcement Learning task (RL, Sutton & Barto, 1998) into several simpler ones. They generally follow a method where agents are trained independently and generally greedily to their local optimality, and are aggregated into a global policy by voting or averaging. Recent works (Jaderberg et al., 2016; van Seijen et al., 2017b) prove their ability to solve problems that are intractable otherwise. Section 2 provides a survey of approaches and algorithms in this field.

Formalised in Section 3, the Multi-Advisor RL (MAd-RL) partitions a single-agent RL into a Multi-Agent RL problem (Shoham et al., 2003), under the widespread *divide & conquer* paradigm. Unlike Hierarchical RL (Dayan & Hinton, 1993; Parr & Russell, 1998; Dietterich, 2000a), this approach gives them the role of advisor: providing an aggregator with the local $Q$-values for all actions. The advisors are said to have a *focus*: reward function, state space, learning technique, etc. The MAd-RL approach allows therefore to tackle the RL task from different focuses.

When a person is consulted for an advice by a enquirer, he may answer egocentrically: as if he was in charge of next actions, agnostically: anticipating any future actions equally, or empathically: by considering the next actions of the enquirer. The same approaches are modelled in the local advisors' planning methods. Section 4 shows that the *egocentric* planning presents the severe theoretical shortcoming of inverting a $\max \sum$ into a $\sum \max$ in the global Bellman equation. It leads to an overestimation of the values of states where the advisors disagree, and creates an *attractor* phenomenon, causing the system to remain static without any tie-breaking possibilities. It is shown on a navigation task that attractors can be avoided by lowering the discount factor $\gamma$ under a given value. The *agnostic* planning (van Seijen et al., 2017b) has the drawback to be inefficient in dangerous environments, because it gets easily afraid of the controller performing a bad sequence of actions. Finally, we introduce our novel *empathic* planning and show that it converges to the global optimal Bellman equation when all advisors are training on the full state space.

van Seijen et al. (2017a) demonstrate on a fruit collection task that a distributed architecture significantly speeds up learning and converges to a better solution than non distributed baselines. Section 5.2 extends those results and empirically validates our theoretical analysis: the *egocentric* planning gets stuck in attractors with high $\gamma$ values; with low $\gamma$ values, it gets high scores but is also very unstable as soon as some noise is introduced; the *agnostic* planning fails at efficiently gathering the fruits near the ghosts; despite lack of convergence guarantees with partial information in advisors' state space, our novel *empathic* planning also achieves high scores while being robust to noise.

## 2    RELATED WORK

**Task decomposition** – Literature features numerous ways to distribute a single-agent RL problem over several specialised advisors: state space approximation/reduction (Böhmer et al., 2015), reward segmentation (Dayan & Hinton, 1993; Gábor et al., 1998; Vezhnevets et al., 2017; van Seijen et al., 2017b), algorithm diversification (Wiering & Van Hasselt, 2008; Laroche & Féraud, 2017), algorithm randomization (Breiman, 1996; Glorot & Bengio, 2010), sequencing of actions (Sutton et al., 1999), or factorisation of actions (Laroche et al., 2009). In this paper, we mainly focus on reward segmentation and state space reduction but the findings are applicable to any family of advisors.

**Subtasks aggregation** – Singh & Cohn (1998) are the first to propose to merge Markov decision Processes through their value functions. It makes the following strong assumptions: positive rewards, model-based RL, and local optimality is supposed to be known. Finally, the algorithm simply accompanies a classical RL algorithm by pruning actions that are known to be suboptimal. Sprague & Ballard (2003) propose to use a local SARSA online learning algorithm for training the advisors, but they elude the fact that the online policy cannot be locally accurately estimated with partial state space, and that this endangers the convergence properties. Russell & Zimdars (2003) study more in depth the theoretical guaranties of convergence to optimality with the local $Q$-learning, and the local SARSA algorithms. However, their work is limited in the fact that they do not allow the local advisors to be trained on local state space. van Seijen et al. (2017b) relax this assumption at the expense of optimality guarantees and beat one of the hardest Atari games: Ms. Pac-Man, by decomposing the task into hundreds of subtasks that are trained in parallel.

MAd-RL can also be interpreted as a generalisation of Ensemble Learning (Dietterich, 2000b) for RL. As such, Sun & Peterson (1999) use a boosting algorithm in a RL framework, but the boosting is performed upon policies, not RL algorithms. In this sense, this article can be seen as a precursor to the policy reuse algorithm (Fernández & Veloso, 2006) rather than a multi-advisor framework. Wiering & Van Hasselt (2008) combine five online RL algorithms on several simple RL problems and show that some mixture models of the five experts performs generally better than any single one alone. Each algorithm tackles the whole task. Their algorithms were off-policy, on-policy, actor-critics, etc. Faußer & Schwenker (2011) continue this effort in a very specific setting where actions are explicit and deterministic transitions. We show in Section 4 that the planning method choice is critical and that some recommendations can be made in accordance to the task definition. In Harutyunyan et al. (2015), while all advisors are trained on different reward functions, these are potential based reward shaping variants of the same reward function. They are therefore embedding the same goals. As a consequence, it can be related to a bagging procedure. The advisors recommendation are then aggregated under the HORDE architecture (Sutton et al., 2011), with *egocentric* planning. Two aggregator functions are tried out: majority voting and ranked voting. Laroche & Féraud (2017) follow a different approach in which, instead of boosting the weak advisors performances by aggregating their recommendation, they select the best advisor. This approach is beneficial for staggered learning, or when one or several advisors may not find good policies, but not for variance reduction brought by the committee, and it does not apply to compositional RL.

The UNREAL architecture (Jaderberg et al., 2016) improves the state-of-the art on Atari and Labyrinth domains by training their deep network on auxiliary tasks. They do it in an unsupervised manner and do not consider each learner as a direct contributor to the main task. The bootstrapped DQN architecture Osband et al. (2016) also exploits the idea of multiplying the $Q$-value estimations to favour deep exploration. As a result, UNREAL and bootstrapped DQN do not allow to break down a task into smaller, tractable pieces.

As a summary, a large variety of papers are published on these subjects, differing by the way they factorise the task into subtasks. Theoretical obstacles are identified in Singh & Cohn (1998) and Russell & Zimdars (2003), but their analysis does not go further than the non-optimality observation in the general case. In this article, we accept the non-optimality of the approach, because it naturally comes from the simplification of the task brought by the decomposition and we analyse the pros and cons of each planning methods encountered in the literature. But first, Section 3 lays the theoretical foundation for Multi-Advisor RL.

**Domains** – Related works apply their distributed models to diverse domains: racing (Russell & Zimdars, 2003), scheduling (Russell & Zimdars, 2003), dialogue (Laroche & Féraud, 2017), and fruit collection (Singh & Cohn, 1998; Sprague & Ballard, 2003; van Seijen et al., 2017a;b).

The fruit collection task being at the centre of attention, it is natural that we empirically validate our theoretical findings on this domain: the Pac-Boy game (see Figure 1), borrowed from van Seijen et al. (2017a). Pac-Boy navigates in a 11x11 maze with a total of 76 possible positions and 4 possible actions in each state: $\mathcal{A} = \{N, W, S, E\}$, respectively for North, West, South and East. Bumping into a wall simply causes the player not to move without penalty. Since Pac-Boy always starts in the same position, there are 75 potential fruit positions. The fruit distribution is randomised: at the start of each new episode, there is a 50% probability for each position to have a fruit. A game lasts until the last fruit has been eaten, or after the $300^{\text{th}}$ time step. During an episode, fruits remain fixed until they get eaten by Pac-Boy. Two randomly-moving ghosts are preventing Pac-Boy from eating all the fruits. The state of the game consists of the positions of Pac-Boy, fruits, and ghosts: $76 \times 2^{75} \times 76^2 \approx 10^{28}$ states. Hence, no global representation system can be implemented without using function approximation. Pac-Boy gets a $+1$ reward for every eaten fruit and a $-10$ penalty when it is touched by a ghost.

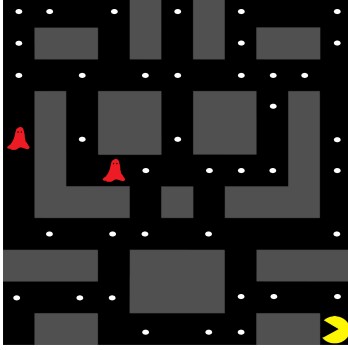

Figure 1: The Pac-Boy game: Pac-Boy is yellow, the corridors are in black, the walls in grey, the fruits are the white dots and the ghosts are in red.

## 3 MULTI-ADVISOR REINFORCEMENT LEARNING

**Markov Decision Process** – The Reinforcement Learning (RL) framework is formalised as a Markov Decision Process (MDP). An MDP is a tuple $\langle \mathcal{X}, \mathcal{A}, P, R, \gamma \rangle$ where $\mathcal{X}$ is the state space, $\mathcal{A}$ is the action space, $P : \mathcal{X} \times \mathcal{A} \to \mathcal{X}$ is the Markovian transition stochastic function, $R : \mathcal{X} \times \mathcal{A} \to \mathbb{R}$ is the immediate reward stochastic function, and $\gamma$ is the discount factor.

A *trajectory* $\langle x(t), a(t), x(t+1), r(t) \rangle_{t \in [\![0, T-1]\!]}$ is the projection into the MDP of the task episode. The goal is to generate trajectories with high discounted cumulative reward, also called more succinctly *return*: $\sum_{t=0}^{T-1} \gamma^t r(t)$. To do so, one needs to find a policy $\pi : \mathcal{X} \times \mathcal{A} \to [0, 1]$ maximising the $Q$-function: $Q_\pi(x, a) = \mathbb{E}_\pi \left[ \sum_{t' \geq t} \gamma^{t'-t} R(X_{t'}, A_{t'}) | X_t = x, A_t = a \right]$.

**MAd-RL structure** – This section defines the Multi-Advisor RL (MAd-RL) framework for solving a single-agent RL problem. The $n$ advisors are regarded as specialised, possibly weak, learners that are concerned with a sub part of the problem. Then, an aggregator is responsible for merging the advisors' recommendations into a global policy. The overall architecture is illustrated in Figure 2. At each time step, an advisor $j$ sends to the aggregator its local $Q$-values for all actions in the current state.

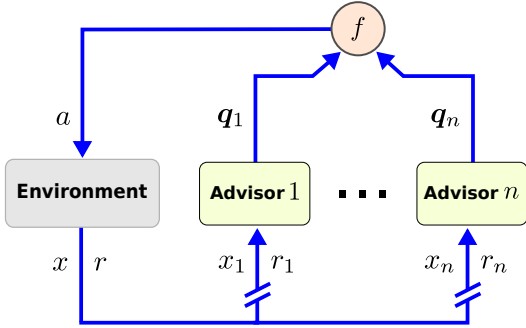

Figure 2: The MAd-RL architecture

**Aggregating advisors' recommendations** – In Figure 2, the $f$ function's role is to aggregate the advisors' recommendations into a policy. More formally, the aggregator is defined with $f : \mathbb{R}^{n \times |\mathcal{A}|} \to \mathcal{A}$, which maps the received $\boldsymbol{q}_j = Q_j(x_j, \cdot)$ values into an action of $\mathcal{A}$. This article focuses on the analysis of the way the local $Q_j$-functions are computed. From the values $\boldsymbol{q}_j$, one can design a $f$ function that implements any aggregator function encountered in the Ensemble methods literature (Dietterich, 2000b): voting schemes (Gibbard, 1973), Boltzmann policy mixtures (Wiering & Van Hasselt, 2008) and of course linear value-function combinations (Sun & Peterson, 1999; Russell & Zimdars, 2003; van Seijen et al., 2017b). For the further analysis, we restrict ourselves to the linear decomposition of the rewards: $R(x, a) = \sum_j w_j R_j(x_j, a)$, which implies the same decomposition of return if they share the same $\gamma$. The advisor's state representation may be locally defined by $\phi_j : \mathcal{X} \to \mathcal{X}_j$, and its local state is denoted by $x_j = \phi_j(x) \in \mathcal{X}_j$. We define the aggregator function $f_\Sigma(x)$ as being greedy over the $Q_j$-functions aggregation $Q_\Sigma(x, a)$:

$$Q_\Sigma(x,a) = \sum_j w_j Q_j(x_j, a),$$

$$f_\Sigma(x) = \operatorname*{argmax}_{a \in \mathcal{A}} Q_\Sigma(x,a).$$

We recall hereunder the main theoretical result of van Seijen et al. (2017b): a theorem ensuring, under conditions, that the advisors' training eventually converges. Note that by assigning a stationary behaviour to each of the advisors, the sequence of random variables $X_0, X_1, X_2, \ldots$, with $X_t \in \mathcal{X}$ is a Markov chain. For later analysis, we assume the following.

**Assumption 1.** *All the advisors' environments are Markov:*

$$\mathbb{P}(X_{j,t+1}|X_{j,t}, A_t) = \mathbb{P}(X_{j,t+1}|X_{j,t}, A_t, \ldots, X_{j,0}, A_0).$$

**Theorem 1** (van Seijen et al. (2017b)). *Under Assumption 1 and given any fixed aggregator, global convergence occurs if all advisors use off-policy algorithms that converge in the single-agent setting.*

Although Theorem 1 guarantees convergence, it does not guarantee the optimality of the converged solution. Moreover, this fixed point only depends on each advisor model and on their planning methods (see Section 4), but not on the particular optimisation algorithms that are used by them.

## 4 ADVISORS' PLANNING METHODS

This section present three planning methods at the advisor's level. They differ in the policy they evaluate: *egocentric* planning evaluates the local greedy policy, *agnostic* planning evaluates the random policy, and *empathic* planning evaluates the aggregator's greedy policy.

### 4.1 *Egocentric* PLANNING

The most common approach in the literature is to learn off-policy by bootstrapping on the locally greedy action: the advisor evaluates the local greedy policy. This planning, referred to in this paper as *egocentric*, has already been employed in Singh & Cohn (1998), Russell & Zimdars (2003), Harutyunyan et al. (2015), and van Seijen et al. (2017a). Theorem 1 guarantees for each advisor $j$ the convergence to the local optimal value function, denoted by $Q_j^{ego}$, which satisfies the Bellman optimality equation:

$$Q_j^{ego}(x_j, a) = \mathbb{E}\left[r_j + \gamma \max_{a' \in \mathcal{A}} Q_j^{ego}(x_j', a')\right],$$

where the local immediate reward $r_j$ is sampled according to $R_j(x_j, a)$, and the next local state $x_j'$ is sampled according to $P_j(x_j, a)$. In the aggregator global view, we get:

$$Q_\Sigma^{ego}(x,a) = \sum_j w_j Q_j^{ego}(x_j, a) = \mathbb{E}\left[\sum_j w_j r_j + \gamma \sum_j w_j \max_{a' \in \mathcal{A}} Q_j^{ego}(x_j', a')\right].$$

By construction, $r = \sum_j w_j r_j$, and therefore we get:

$$Q_\Sigma^{ego}(x,a) \geq \mathbb{E}\left[r + \gamma \max_{a' \in \mathcal{A}} \sum_j w_j Q_j^{ego}(x_j', a')\right]$$

$$\geq \mathbb{E}\left[r + \gamma \max_{a' \in \mathcal{A}} Q_\Sigma^{ego}(x', a')\right].$$

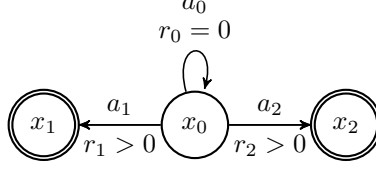

Figure 3: Attractor example.

*Egocentric* planning suffers from an inversion between the $\max$ and $\sum$ operators and, as a consequence, it overestimate the state-action values when the advisors disagree on the optimal action. This flaw has critical consequences in practice: it creates *attractor* situations. Before we define and

study them formally, let us explain attractors with an illustrative example based on the simple MDP depicted in Figure 3. In initial state $x_0$, the system has three possible actions: stay put (action $a_0$), perform advisor 1's goal (action $a_1$), or perform advisor 2's goal (action $a_2$). Once achieving a goal, the trajectory ends. The $Q$-values for each action are easy to compute: $Q_\Sigma^{ego}(s, a_0) = \gamma r_1 + \gamma r_2$, $Q_\Sigma^{ego}(s, a_1) = r_1$, and $Q_\Sigma^{ego}(s, a_2) = r_2$.

As a consequence, if $\gamma > r_1/(r_1 + r_2)$ and $\gamma > r_2/(r_1 + r_2)$, the local *egocentric* planning commands to execute action $a_0$ *sine die*. This may have some apparent similarity with the Buridan's ass paradox (Zbilut, 2004; Rescher, 2005): a donkey is equally thirsty and hungry and cannot decide to eat or to drink and dies of its inability to make a decision. The determinism of judgement is identified as the source of the problem in antic philosophy. Nevertheless, the *egocentric* sub-optimality does not come from actions that are equally good, nor from the determinism of the policy, since adding randomness to the system will not help. Let us define more generally the concept of attractors.

**Definition 1.** *An attractor $x$ is a state where the following strict inequality holds:*

$$\max_{a \in \mathcal{A}} \sum_j w_j Q_j^{ego}(x_j, a) < \gamma \sum_j w_j \max_{a \in \mathcal{A}} Q_j^{ego}(x_j, a).$$

**Theorem 2.** *State $x$ is attractor, if and only if the optimal egocentric policy is to stay in $x$ if possible.*

Note that there is no condition in Theorem 2 (proved in appendix, Section A) on the existence of actions allowing the system to be actually static. Indeed, the system might be stuck in an attractor set, keep moving, but opt to never achieve its goals. To understand how this may happen, just replace state $x_0$ in Figure 3 with an attractor set of similar states: where action $a_0$ performs a random transition in the attractor set, and actions $a_1$ and $a_2$ respectively achieve tasks of advisors 1 and 2. Also, it may happen that an attractor set is escapable by the lack of actions keeping the system in an attractor set. For instance, in Figure 3, if action $a_0$ is not available, $x_0$ remains an attractor, but an unstable one.

**Definition 2.** *An advisor $j$ is said to be progressive if the following condition is satisfied:*
$$\forall x_j \in \mathcal{X}_j, \forall a \in \mathcal{A}, \quad Q_j^{ego}(x_j, a) \geq \gamma \max_{a' \in \mathcal{A}} Q_j^{ego}(x_j, a').$$

The intuition behind the progressive property is that no action is worse than losing one turn to do nothing. In other words, only progress can be made towards this task, and therefore non-progressing actions are regarded by this advisor as the worst possible ones.

**Theorem 3.** *If all the advisors are progressive, there cannot be any attractor.*

The condition stated in Theorem 3 (proved in appendix, Section A) is very restrictive. Still, there exist some RL problems where Theorem 3 can be applied, such as resource scheduling where each advisor is responsible for the progression of a given task. Note that a MAd-RL setting without any attractors does not guarantee optimality for the *egocentric* planning. Most of RL problems do not fall into this category. Theorem 3 neither applies to RL problems with states that terminate the trajectory while some goals are still incomplete, nor to navigation tasks: when the system goes into a direction that is opposite to some goal, it gets into a state that is worse than staying in the same position.

**Navigation problem attractors** – We consider the three-fruit attractor illustrated in Figure 4: moving towards a fruit, makes it closer to one of the fruits, but further from the two other fruits (diagonal moves are not allowed). The expression each action $Q$-value is as follows: $Q_\Sigma^{ego}(x, S) = \gamma \sum_j \max_{a \in \mathcal{A}} Q_j^{ego}(x_j, a) = 3\gamma^2$, and $Q_\Sigma^{ego}(x, N) = Q_\Sigma^{ego}(x, E) = Q_\Sigma^{ego}(x, W) = \gamma + 2\gamma^3$. That means that, if $\gamma > 0.5$, $Q_\Sigma^{ego}(x, S) > Q_\Sigma^{ego}(x, N) = Q_\Sigma^{ego}(x, E) = Q_\Sigma^{ego}(x, W)$. As a result, the aggregator would opt to go South and hit the wall indefinitely.

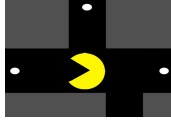

Figure 4: 3-fruit attractor.

More generally in a deterministic[1] task where each action $a$ in a state $x$ can be cancelled by a new action $a_x^{-1}$, it can be shown that the condition on $\gamma$ is a function of the size of the action set $\mathcal{A}$. Theorem 4 is proved in Section A of the appendix.

**Theorem 4.** *State $x \in \mathcal{X}$ is guaranteed not to be an attractor if $\forall a \in \mathcal{A}, \exists a_x^{-1} \in \mathcal{A}$, such that*

$$P(P(x, a), a_x^{-1}) = x \text{ , if } \forall a \in \mathcal{A}, R(x, a) \geq 0 \text{ , and if } \gamma \leq \frac{1}{|\mathcal{A}| - 1} \text{ .}$$

---

[1] A more general result on stochastic navigation tasks can be demonstrated. We limited the proof to the deterministic case for the sake of simplicity.

### 4.2 *Agnostic* PLANNING

The *agnostic* planning does not make any prior on future actions and therefore evaluates the random policy. Once again, Theorem 1 guarantees the convergence of the local optimisation process to its local optimal value, denoted by $Q_j^{agn}$, which satisfies the following Bellman equation:

$$Q_j^{agn}(x_j, a) = \mathbb{E}\left[r_j + \frac{\gamma}{|\mathcal{A}|}\sum_{a' \in \mathcal{A}} Q_j^{agn}(x_j', a')\right],$$

$$Q_{\Sigma}^{agn}(x, a) = \mathbb{E}\left[r + \frac{\gamma}{|\mathcal{A}|}\sum_j w_j \sum_{a' \in \mathcal{A}} Q_j^{agn}(x_j', a')\right]$$

$$= \mathbb{E}\left[r + \frac{\gamma}{|\mathcal{A}|}\sum_{a' \in \mathcal{A}} Q_{\Sigma}^{agn}(x', a')\right].$$

Local *agnostic* planning is equivalent to the global *agnostic* planning. Additionally, as opposed to the *egocentric* case, r.h.s. of the above equation does not suffer from $\max$-$\sum$ inversion. It then follows that no attractor are present in *agnostic* planning. Nevertheless, acting greedily with respect to $Q_{\Sigma}^{agn}(x, a)$ only guarantees to be better than a random policy and in general may be far from being optimal. Still, *agnostic* planning has proved its usefulness on Ms. Pac-Man (van Seijen et al., 2017b).

### 4.3 *Empathic* PLANNING

A novel approach, inspired from the online algorithm found in Sprague & Ballard (2003); Russell & Zimdars (2003) is to locally predict the aggregator's policy. In this method, referred to as *empathic*, the aggregator is in control, and the advisors are evaluating the current aggregator's greedy policy $f$ with respect to their local focus. More formally, the local Bellman equilibrium equation is the following one:

$$Q_j^{ap}(x_j, a) = \mathbb{E}\left[r_j + \gamma Q_j^{ap}(x_j', f_{\Sigma}(x'))\right].$$

**Theorem 5.** *Assuming that all advisors are defined on the full state space, MAd-RL with empathic planning converges to the global optimal policy.*

*Proof.*

$$Q_{\Sigma}^{ap}(x, a) = \mathbb{E}\left[r + \gamma \sum_j w_j Q_j^{ap}(x_j', f_{\Sigma}(x'))\right]$$

$$= \mathbb{E}\left[r + \gamma Q_{\Sigma}^{ap}(x', f_{\Sigma}(x'))\right]$$

$$= \mathbb{E}\left[r + \gamma Q_{\Sigma}^{ap}(x', \underset{a' \in \mathcal{A}}{\arg\max} Q_{\Sigma}^{ap}(x', a'))\right]$$

$$= \mathbb{E}\left[r + \gamma \underset{a' \in \mathcal{A}}{\max} Q_{\Sigma}^{ap}(x', a')\right].$$

$Q_{\Sigma}^{ap}$ is the unique solution to the global Bellman optimality equation, and therefore equals the optimal value function, *quod erat demonstrandum.* □

However, most MAd-RL settings involve taking advantage of state space reduction to speed up learning, and in this case, there is no guarantee of convergence because function $f_{\Sigma}(x')$ can only be approximated from the local state space scope. As a result the local estimate $\hat{f}_j(x')$ is used instead of $f_{\Sigma}(x')$ and the reconstruction of $\max_{a' \in \mathcal{A}} Q_{\Sigma}^{ap}(x', a')$ is not longer possible in the global Bellman equation:

$$Q_j^{ap}(x_j, a) = \mathbb{E}\left[r_j + \gamma Q_j^{ap}(x_j', \hat{f}_j(x'))\right],$$

$$Q_{\Sigma}^{ap}(x, a) = \mathbb{E}\left[r + \gamma \sum_j w_j Q_j^{ap}(x_j', \hat{f}_j(x'))\right].$$

## 5 EXPERIMENTS

### 5.1 VALUE FUNCTION APPROXIMATION

For this experiment, we intend to show that the value function is easier to learn with the MAd-RL architecture. We consider a fruit collection task where the agent has to navigate through a $5 \times 5$ grid and receives a +1 reward when visiting a fruit cell (5 are randomly placed at the beginning of each episode). A deep neural network (DNN) is fitted to the ground-truth value function $V_\gamma^{\pi^*}$ for various objective functions: TSP is the optimal number of turns to gather all the fruits, RL is the optimal return, and *egocentric* is the optimal MAd-RL return. This learning problem is fully supervised on 1000 samples, allowing us to show how fast a DNN can capture $V_\gamma^{\pi^*}$ while ignoring the burden of finding the optimal policy and estimating its value functions through TD-backups or value iteration. To evaluate the DNN's performance, actions are selected greedily by moving the agent up, down, left, or right to the neighbouring grid cell of highest value. Section B.1 of the appendix gives the details.

Figure 5a displays the performance of the theoretical optimal policy for each objective function in dashed lines. Here TSP and RL targets largely surpass MAd-RL one. But Figure 5a also displays the performances of the networks trained on the limited data of 1000 samples, for which the results are completely different. The TSP objective target is the hardest to train on. The RL objective target follows as the second hardest to train on. The *egocentric* planning MAd-RL objective is easier to train on, even without any state space reduction, or even without any reward/return decomposition (summed version). Additionally, if the target value is decomposed (vector version), the training is further accelerated. Finally, we found that the MAd-RL performance tends to dramatically decrease when $\gamma$ gets close to 1, because of attractors' presence. We consider this small experiment to show that the complexity of objective function is critical and that decomposing it in the fashion of MAd-RL may make it simpler and therefore easier to train, even without any state space reduction.

### 5.2 PAC-BOY DOMAIN

In this section, we empirically validate the findings of Section 4 in the Pac-Boy domain, presented in Section 2. The MAd-RL settings are associating one advisor per potential fruit location. The local state space consists in the agent position and the existence –or not– of the fruit. Six different settings are compared: the two baselines *linear Q-learning* and *DQN-clipped*, and four MAd-RL settings: *egocentric* with $\gamma = 0.4$, *egocentric* with $\gamma = 0.9$, *agnostic* with $\gamma = 0.9$, and *empathic* with $\gamma = 0.9$. The implementation and experimentation details are available in the appendix, at Section B.2.

We provide links to 5 video files (click on the blue links) representing a trajectory generated at the 50[th] epoch for various settings. *egocentric-$\gamma = 0.4$* adopts a near optimal policy coming close to the ghosts without taking any risk. The fruit collection problem is similar to the travelling salesman problem, which is known to be NP-complete (Papadimitriou, 1977). However, the suboptimal small-$\gamma$ policy consisting of moving towards the closest fruits is in fact a near optimal one. Regarding the ghost avoidance, *egocentric* with small $\gamma$ gets an advantage over other settings: the local optimisation guarantees a perfect control of the system near the ghosts. The most interesting outcome is the presence of the attractor phenomenon in *egocentric-$\gamma = 0.9$*: Pac-Boy goes straight to the centre area of the grid and does not move until a ghost comes too close, which it still knows to avoid perfectly. This is the empirical confirmation that the attractors, studied in Section 4.1, present a real practical issue. *empathic* is almost as good as *egocentric-$\gamma = 0.4$*. *agnostic* proves to be unable to reliably finish the last fruits because it is overwhelmed by the fear of the ghosts, even when they are still far away. This feature of the *agnostic* planning led van Seijen et al. (2017b) to use a dynamic normalisation depending on the number of fruits left on the board. Finally, we observe that *DQN-clipped* also struggles to eat the last fruits.

The quantitative analysis displayed in Figure 5b confirms our qualitative video-based impressions. *egocentric-$\gamma = 0.9$* barely performs better than *linear Q-learning*, *DQN-clipped* is still far from the optimal performance, and gets hit by ghosts from time to time. *agnostic* is closer but only rarely eats all the fruits. Finally, *egocentric-$\gamma = 0.4$* and *empathic* are near-optimal. Only *egocentric-$\gamma = 0.4$* trains a bit faster, and tends to finish the game 20 turns faster too (not shown).

**Results with noisy rewards** – Using a very small $\gamma$ may distort the objective function and perhaps even more importantly the reward signal diminishes exponentially as a function of the distance to the

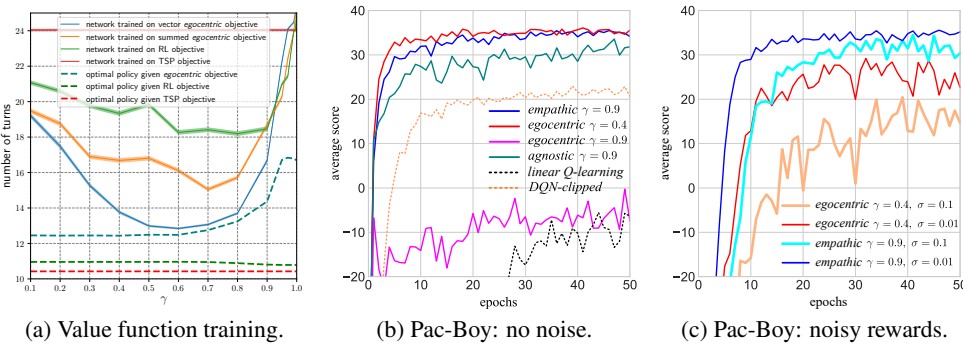

(a) Value function training.    (b) Pac-Boy: no noise.    (c) Pac-Boy: noisy rewards.

Figure 5: Experiment results.

goal, which might have critical consequences in noisy environments, hence the following experiment: several levels of Gaussian centred white noise $\eta_\sigma$ with standard deviation $\sigma \in \{0.01, 0.1\}$ have been applied to the reward signal: at each turn, each advisor now receives $\hat{r}_j = r_j + \eta_\sigma$ instead. Since the noise is centred and white, the ground truth $Q$-functions remain the same, but their estimators obtained during sampling is corrupted by noise variance.

Empirical results displayed in Figure 5c shows that the *empathic* planning performs better than the *egocentric* one even under noise with a 100 times larger variance. Indeed, because of the noise, the fruit advisors are only able to consistently perceive the fruits that are in a radius dependent on $\gamma$ and $\sigma$. The *egocentric* planning, incompatible with high $\gamma$ values, is therefore myopic and cannot perceive distant fruits. The same kind of limitations are expected to be encountered for small $\gamma$ values when the local advisors rely on state approximations, and/or when the transitions are stochastic. This also supports the superiority of the empathic planning in the general case.

## 6    CONCLUSION AND DISCUSSION

This article presented MAd-RL, a common ground for the many recent and successful works decomposing a single-agent RL problem into simpler problems tackled by independent learners. It focuses more specifically on the local planning performed by the advisors. Three of them – two found in the literature and one novel – are discussed, analysed and empirically compared: *egocentric*, *agnostic*, and *empathic*. The lessons to be learnt from the article are the following ones.

The *egocentric* planning has convergence guarantees but overestimates the values of states where the advisors disagree. As a consequence, it suffers from *attractors*: states where the *no-op* action is preferred to actions making progress on a subset of subtasks. Some domains, such as resource scheduling, are identified as attractor-free, and some other domains, such as navigation, are set conditions on $\gamma$ to guarantee the absence of attractor. It is necessary to recall that an attractor-free setting means that the system will continue making progress towards goals as long as there are any opportunity to do so, not that the *egocentric* MAd-RL system will converge to the optimal solution.

The *agnostic* planning also has convergence guarantees, and the local *agnostic* planning is equivalent to the global *agnostic* planning. However, it may converge to bad solutions. For instance, in dangerous environments, it considers all actions equally likely, it favours staying away from situation where a random sequence of actions has a significant chance of ending bad: crossing a bridge would be avoided. Still, the *agnostic* planning simplicity enables the use of general value functions (Sutton et al., 2011) as in van Seijen et al. (2017b).

The *empathic* planning optimises the system according to the global Bellman optimality equation, but without any guarantee of convergence, if the advisor state space is smaller than the global state. In our experiments, we never encountered a case where the convergence was not obtained, and on the Pac-Boy domain, it robustly learns a near optimal policy after only 10 epochs. It can also be safely applied to Ensemble RL tasks where all learners are given the full state space.

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

## A    THEOREMS AND THEIR PROOFS

**Theorem 1** (van Seijen et al. (2017b)). *Under Assumption 1 and given any fixed aggregator, global convergence occurs if all advisors use off-policy algorithms that converge in the single-agent setting.*

*Proof.* Each advisor can be seen as an independent learner training from trajectories controlled by an arbitrary behavioural policy. If Assumption 1 holds, each advisor's environment is Markov and off-policy algorithms can be applied with convergence guarantees. ☐

**Theorem 2.** *State $x$ is attractor, if and only if the optimal egocentric policy is to stay in $x$ if possible.*

*Proof.* The logical biconditional will be demonstrated by successively proving the two converse conditionals.

First, the sufficient condition: let us assume that state $x$ is an attractor. By Definition 1, if state $x$ is an attractor, we have:

$$\max_{a \in \mathcal{A}} \sum_j w_j Q_j^{ego}(x_j, a) < \gamma \sum_j w_j \max_{a \in \mathcal{A}} Q_j^{ego}(x_j, a).$$

Let $a_0$ denote the potential action to stay in $x$, and consider the MDP augmented with $a_0$ in $x$. Then, we have :

$$Q_j^{ego}(x, a_0) = \gamma \sum_j w_j \max_{a \in \mathcal{A}} Q_j^{ego}(x_j, a)$$

$$> \max_{a \in \mathcal{A}} \sum_j w_j Q_j^{ego}(x_j, a)$$

$$= \max_{a \in \mathcal{A}} Q_\Sigma^{ego}(x, a),$$

which proves that $a_0$, if possible, will be preferred to any other action $a \in \mathcal{A}$.

Second, the reciprocal condition: let us assume that, in state $x$, action $a_0$ would be preferred by an optimal policy under the *egocentric* planning. Then:

$$Q_j^{ego}(x, a_0) > \max_{a \in \mathcal{A}} Q_\Sigma^{ego}(x, a)$$

$$\gamma \sum_j w_j \max_{a \in \mathcal{A}} Q_j^{ego}(x_j, a) > \max_{a \in \mathcal{A}} \sum_j w_j Q_j^{ego}(x_j, a),$$

which proves that $x$ is an attractor. ☐

**Theorem 3.** *If all the advisors are progressive, there cannot be any attractor.*

*Proof.* Let sum Definition 2 over advisors:

$$\sum_j w_j Q_j^{ego}(x_j, a) \geq \gamma \sum_j w_j \max_{a' \in \mathcal{A}} Q_j^{ego}(x_j, a'),$$

$$\max_{a' \in \mathcal{A}} \sum_j w_j Q_j^{ego}(x_j, a') \geq \sum_j w_j Q_j^{ego}(x_j, a),$$

which proves the theorem. ☐

**Theorem 4.** *State $x \in \mathcal{X}$ is guaranteed not to be an attractor if:*

- $\forall a \in \mathcal{A}, \exists a_x^{-1} \in \mathcal{A}$, *such that* $P(P(x, a), a_x^{-1}) = x$ ,

- $\forall a \in \mathcal{A}, R(x, a) \geq 0$ ,

- $\gamma \leq \dfrac{1}{|\mathcal{A}| - 1}$ .

*Proof.* Let us denote $\mathcal{J}_a^x$ as the set of advisors for which action $a$ is optimal in state $x$. $Q_a^{ego}(x)$ is defined as the sum of perceived value of performing $a$ in state $x$ by the advisors that would choose it:

$$Q_a^{ego}(x) = \sum_{j \in \mathcal{J}_a^x} w_j Q_j^{ego}(x_j', a).$$

Let $a^+$ be the action that maximises this $Q_a^{ego}(x)$ function:

$$a^+ = \operatorname*{argmax}_{a \in \mathcal{A}} Q_a^{ego}(x).$$

We now consider the left hand side of the inequality characterising the attractors in Definition 1:

$$\max_{a \in \mathcal{A}} \sum_j w_j Q_j^{ego}(x_j, a) \geq \sum_j w_j Q_j^{ego}(x_j, a^+),$$

$$= Q_{a^+}^{ego}(x) + \sum_{j \notin \mathcal{J}_{a^+}^x} w_j Q_j^{ego}(x_j, a^+),$$

$$= Q_{a^+}^{ego}(x) + \sum_{j \notin \mathcal{J}_{a^+}^x} w_j \left( R(x, a^+) + \gamma \max_{a' \in \mathcal{A}} Q_j^{ego}(x_j', a') \right).$$

Since $R(x, a^+) \geq 0$, and since the $a'$ maximising $Q_j^{ego}(x_j', a')$ is at least as good as the cancelling action $(a^+)_x^{-1}$, we can follow with:

$$\max_{a \in \mathcal{A}} \sum_j w_j Q_j^{ego}(x_j, a) \geq Q_{a^+}^{ego}(x) + \sum_{j \notin \mathcal{J}_{a^+}^x} w_j \gamma^2 \max_{a \in \mathcal{A}} Q_j^{ego}(x_j, a).$$

By comparing this last result with the right hand side of Definition 1, the condition for $x$ not being an attractor becomes:

$$(1 - \gamma) Q_{a^+}^{ego}(x) \geq (1 - \gamma) \gamma \sum_{j \notin \mathcal{J}_{a^+}^x} w_j \max_{a \in \mathcal{A}} Q_j^{ego}(x_j, a),$$

$$Q_{a^+}^{ego}(x) \geq \gamma \sum_{a \neq a^+} \sum_{j \in \mathcal{J}_a^x} w_j Q_j^{ego}(x_j, a),$$

$$Q_{a^+}^{ego}(x) \geq \gamma \sum_{a \neq a^+} Q_a^{ego}(x).$$

It follows directly from the inequality $Q_{a^+}^{ego}(x) \geq Q_a^{ego}(x)$, that $\gamma \leq 1/(|\mathcal{A}|-1)$ guarantees the absence of attractor. $\qquad\square$

# B  Experimental details

## B.1  Value function approximation

All trainings are performed from the same state and the same network, which are described in Section B.1.1. Their only difference lies in the objective function targets that are detailed in Section B.1.2.

### B.1.1  Neural network setting

Similarly to the Taxi Domain Dieterich (1998), we incorporate the location of the fruits into the state representation by using a 50 dimensional bit vector, where the first 25 entries are used for fruit positions, and the last 25 entries are used for the agent's position. The DNN feeds this bit-vector as the input layer into two dense hidden layers with 100 and then 50 units. The output is a single linear head representing the state-value, or a multiple in the case of the vector MAd-RL target. In order to assess the value function complexity, we train for each discount factor setting a DNN of fixed size on 1000 random states with their ground truth values. Each DNN is trained over 500 epochs using the Adam optimizer Kingma & Ba (2014) with default parameters.

### B.1.2  Objective function targets

Four different objective function targets are considered:

- The TSP objective function target is the natural objective function, as defined by the Travelling Salesman Problem: the number of turns to gather all the fruits:

$$y_{TSP}(x) = - \min_{\sigma \in \Sigma_k} \left[ \sum_{i=1}^{k} d(x_{\sigma(i-1)}, x_{\sigma(i)}) \right],$$

  where $k$ is the number of fruits remaining in state $x$, where $\Sigma_k$ is the ensemble of all permutations of integers between 1 and $k$, where $\sigma$ is one of those permutations, where $x_0$ is the position of the agent in $x$, where $x_i$ for $1 \le i \le k$ is the position of fruit with index $i$, where $d(x_i, x_j)$ is the distance ($||\cdot||_1$ in our gridworld) between positions $x_i$ and $x_j$.

- The RL objective function target is the objective function defined for an RL setting, which depends on the discount factor $\gamma$:

$$y_{RL}(x) = \max_{\sigma \in \Sigma_k} \left[ \sum_{i=1}^{k} \gamma^{\sum_{j=1}^{i} d(x_{\sigma(j-1)}, x_{\sigma(j)})} \right],$$

  with the same notations as for TSP.

- The summed *egocentric* planning MAd-RL objective function target does not involve the search into the set of permutations and can be considered simpler to this extent:

$$y_{ego}(x) = \left[ \sum_{i=1}^{k} \gamma^{d(x_0, x_i)} \right],$$

  with the same notations as for TSP.

- The vector *egocentric* planning MAd-RL objective function target is the same as the summed one, except that the target is now a vector, separated into as many channels as potential fruit position:

$$\boldsymbol{y}_{ego}(x) = \begin{cases} \gamma^{d(x_0, x_i)} & \text{if there is a fruit in } x_i, \\ 0 & \text{otherwise.} \end{cases}$$

## B.2  Pac-Boy experiment

**MAd-RL Setup** – Each advisor is responsible for a specific source of reward (or penalty). More precisely, we assign an advisor to each possible fruit location. This advisor sees a +1 reward only if a fruit at its assigned position gets eaten. Its state space consists of Pac-Boy's position, resulting in 76 states. In addition, we assign an advisor to each ghost. This advisor receives a -10 reward if Pac-Boy

bumps into its assigned ghost. Its state space consists of Pac-Boy's and ghost's positions, resulting in $76^2$ states. A fruit advisor is only active when there is a fruit at its assigned position. Because there are on average 37.5 fruits, the average number of advisors running at the beginning of each episode is 39.5. Each fruit advisor is set inactive when its fruit is eaten.

The learning was performed through Temporal Difference updates. Due to the small state spaces for the advisors, we can use a tabular representation. We train all learners in parallel with off-policy learning, with Bellman residuals computed as presented in Section 4 and a constant $\alpha = 0.1$ parameter. The aggregator function sums the $Q$-values for each action $a \in \mathcal{A}$: $Q_\Sigma(x, a) := \sum_j Q_j(x_j, a)$, and uses $\epsilon$-greedy action selection with respect to these summed values. Because ghost agents have exactly identical MDP, we also benefit from direct knowledge transfer by sharing their $Q$-tables.

One can notice that Assumption 1 holds in this setting and that, as a consequence, Theorem 1 applies for the *egocentric* and *agnostic* planning methods. Theorem 4 determines sufficient conditions for not having any attractor in the MDP. In the Pac-Boy domain, the cancelling action condition is satisfied for every $x \in \mathcal{X}$. As for the $\gamma$ condition, it is not only sufficient but also necessary, since being surrounded by goals of equal value is an attractor if $\gamma > 1/3$. In practice, an attractor becomes stable only when there is an action enabling it to remain in the attraction set. Thus, the condition for not being stuck in an attractor set can be relaxed to $\gamma \leq 1/(|\mathcal{A}|-2)$. Hence, the result of $\gamma > 1/2$ in the example illustrated by Figure 4.

**Baselines** – The first baseline is the standard DQN algorithm (Mnih et al., 2015) with reward clipping (referred to as *DQN-clipped*). Its input is a 4-channel binary image with the following features: the walls, the ghosts, the fruits, or Pac-Boy. The second baseline is a system that uses the exact same input features as the MAd-RL model. Specifically, the state of each advisor of the MAd-RL model is encoded with a one-hot vector and all these vectors are concatenated, resulting in a sparse binary feature vector of size 17, 252. This vector is used for linear function approximation with $Q$-learning. We refer to this setting with *linear Q-learning*. We also tried to train deep architectures from these features with no success.

**Experimental setting** – Time scale is divided into 50 epochs lasting 20,000 transitions each. At the end of each epoch an evaluation phase is launched for 80 games. The theoretical expected maximum score is 37.5 and the random policy average score is around -80.

Explicit links to the videos (www.streamable.com website was used to ensure anonymity, if accepted the videos will be linked to a more sustainable website):

- *egocentric-$\gamma = 0.4$*: https://streamable.com/6tian
- *egocentric-$\gamma = 0.9$*: https://streamable.com/sgjkq
- *empathic*: https://streamable.com/h6gey
- *agnostic*: https://streamable.com/grswh
- *DQN-clipped*: https://streamable.com/emh6y

