# OpenReview forum: "Multi-Advisor Reinforcement Learning"
_ICLR.cc/2018/Conference — Reject_

### Official Review · AnonReviewer3 · 2017-11-17
**The paper presents a somewhat unifying wide-angle view of multi-learner RL, but the paper is unclear and unfocused**

**Rating:** 4
**Confidence:** 4

**Review:**

The paper presents Multi-Advisor RL (MAd-RL), a formalized view of many forms of performing RL by training multiple learners, then aggregating their results into a single decision-making agent.  Previous work and citations are plentiful and complete, and the field of study is a promising approach to RL.  Through MAd-RL, the authors analyze the effects of egocentric, agnostic, and empathic planning at the sub-learner level on the resulting applied aggregated policy.  After this theoretical discussion, the different types of sub-learners are used on a Pac-Man problem.

I believe an interesting paper lies within this, and were this a journal, would recommend edits and resubmission.  However, in its current state, the paper is too disorganized and unclear to merit publication.  It took quite a bit of time for me to understand what the authors wanted me to focus on - the paper needs a clearer statement early summarizing its intended contributions.  In addition, more care to language usage is needed - for example, "an attractor" refers to an MDP in Figure 3, a state in Theorem 2, and a set in the Theorem 2 discussion.  Additionally, the theoretical portion focuses on the effects of the three different sub-learner types, but the experiments are "intend[ed] to show that the value function is easier to learn with the MAd-RL architecture," which is an entirely different goal.

I recommend the authors decide what to focus on, rethink how paper space is allocated, and take care to more clearly drive home their intended point.

---

### Official Review · AnonReviewer1 · 2017-11-27
**Very interesting theoretical analysis. Needs a sharper focus.**

**Rating:** 4
**Confidence:** 4

**Review:**

This paper presents MAd-RL, a method for decomposition of a single-agent RL problem into a simple sub-problems, and aggregating them back together. Specifically, the authors propose a novel local planner - emphatic, and analyze the newly proposed local planner along of two existing ones - egocentric and agnostic. The MAd-RL, and theoretical analysis, is evaluated on the Pac-Boy task, and compared to DQN and Q-learning with function approximation.

Pros:
1. The paper is well written, and well-motivated.
2. The authors did an extraordinary job in building the intuition for the theoretical work, and giving appropriate examples where needed.
3. The theoretical analysis of the paper is extremely interesting. The observation that a linearly weighted reward, implies linearly weighted Q function, analysis of different policies, and local minima that result is the strongest and the most interesting points of this paper.

Cons:
1. The paper is too long. 14 pages total - 4 extra pages (in appendix) over the 8 page limit, and 1 extra page of references. That is 50% overrun in the context, and 100% overrun in the references. The most interesting parts and the most of the contributions are in the Appendix, which makes it hard to assess the contributions of the paper. There are two options:
  1.1 If the paper is to be considered as a whole, the excessive overrun gives this paper unfair advantage over other ICLR papers. The flavor and scope and quality of the problems that can be tackled with 50% more space is substantially different from what can be addressed within the set limit. If the extra space is necessary, perhaps this paper is better suited for another publication?
  1.2 If the paper is assessed only based on the main part without Appendix, then the only novelty is emphatic planner, and the theoretical claims with no proofs. The results are interesting, but are lacking implementation details. Overall, a substandard paper.
2. Experiments are disjoint from the method’s section. For example:
  2.1 Section 5.1 is completely unrelated with the material presented in Section 4.
  2.2 The noise evaluation in Section 5.3 is nice, but not related with the Section 4. This is problematic because, it is not clear if the focus of the paper is on evaluating MAd-RL and performance on the Ms.PacMan task, or experimentally demonstrating claims in Section 4.

Recommendations:
1. Shorten the paper to be within (or close to the recommended length) including Appendix.
2. Focus paper on the analysis of the advisors, and Section 5. on demonstrating the claims.
3. Be more explicit about the contributions.
4. How does the negative reward influence the behavior the agent? The agent receives negative reward when near ghosts.
5. Move the short (or all) proofs from Appendix into the main text.
6. Move implementation details of the experiments (in particular the short ones) into the main text.
7. Use the standard terminology (greedy and random policies vs. egoistic and agnostic) where possible. The new terms for well-established make the paper needlessly more complex.
8. Focus the literature review on the most relevant work, and contrast the proposed work with existing peer reviewed methods.
9. Revise the literature to emphasize more recent peer reviewed references. Only three references are recent (less than 5 years), peer reviewed references, while there are 12 historic references. Try to reduce dependencies on non-peer reviewed references (~10 of them).
10. Make a pass through the paper, and decouple it from the van Seijen et al., 2017a
11. Minor: Some claims need references:
  11.1 Page 5: “egocentric sub-optimality  does not come from the actions that are equally good, nor from the determinism of the policy, since adding randomness…” - Wouldn’t adding epsilon-greediness get the agent unstuck?
  11.2 Page 1. “It is shown on the navigation task ….” - This seems to be shown later in the results, but in the intro it is not clear if some other work, or this one shows it.
12. Minor:
  12.1 Mix genders when talking about people. Don’t assume all people that make “complex and important problems”, or who are “consulted for advice”, are male.
  12.2 Typo: Page 5: a_0 sine die
  12.3 Page 7 - omit results that are not shown
  12.4 Make Figures larger - it is difficult, if not impossible to see
  12.5 What is the difference between Pac-Boy and Ms. Pacman task? And why not use Ms. Packman?

---

### Official Review · AnonReviewer2 · 2017-11-30
**Well-written but lacks deep technical and empirical contributions**

**Rating:** 4
**Confidence:** 5

**Review:**

Summary

The paper is well-written but does not make deep technical contributions and does not present a comprehensive evaluation or highly insightful empirical results.

Abstract / Intro

I get the entire focus of the paper is some variant of Pac-Man which has received attention in the RL literature for Atari games, but for the most part the impressive advances of previous Atari/RL papers are in the setting that the raw video is provided as input, which is much different than solving the underlying clean mathematically abstracted problem (as a grid world with obstacles) as done here and evident in the videos.  Further it is honestly hard for me to be strongly motivated about a paper that focuses on the need to decompose Pac-man into sub-agents/advisor value functions.

Section 2

Another historically well-cited paper for MDP decomposition:

  Flexible Decomposition Algorithms for Weakly Coupled Markov Decision Problems, Ronald Parr. UAI 98.
  https://dslpitt.org/uai/papers/98/p422-parr.pdf

Section 3

Is the additive reward decomposition a required part of the problem specification?  It seems so, i.e., there is no obvious method for automatically decomposing a monolithic reward function over advisors.

Section 4

* Egocentric:

Definition 1: Sure, the problem will have local optima (attractors) when decomposed suboptimally -- I'm not sure what new insight we've gained from this analysis... it is a general problem with any function approximation scheme that does not guarantee that the rank ordering of actions for a state is preserved.

* Agnostic

Other than approximating some type of myopic rollout, I really don't see why this approach would be reasonable?  I am surprised it works at all though my guess is that this could simply be an artifact of evaluating on a single domain with a specific structure.

* Empathic

This appears to be the key contribution though related work certainly infringes on its novelty.  Is this paper then an empirical evaluation of previous methods in a single Pac-man grid world variant?

I wonder if the theory of DEC-MDPs would have any relevance for novel analysis here?

Section 5

I'm disappointed that the authors only evaluate on a single domain; presumably the empathic approach has applications beyond Pac-Man?

The fact that empathic generally performs better is not at all surprising.  The fact that a modified discount factor for egocentric can also perform well is not surprising given that lower discount factors have often been shown to improve approximated MDP solutions, e.g.,

  Biasing Approximate Dynamic Programming with a Lower Discount Factor

  Marek Petrik, Bruno Scherrer (NIPS-08).
  http://marek.petrik.us/pub/Petrik2009a.pdf

***

Side note:

The following part is somewhat orthogonal to the review above in that I would not expect the authors to address this on revision, *but* at the same time I think it provides a connection to the special case of concurrent action decomposition into advisors, which could potentially provide a high impact direction of application for this work (i.e., concurrent problems are hard and show up in numerous operations research problems covering inventory control, logistics, epidemic response).

For the special case that each advisor is assigned to one action in a factored space of concurrent actions, the egocentric algorithm would be very close to the Hindsight approximation in Section 6 of this paper (including an additive decomposition of rewards):

  Planning in Factored Action Spaces with Symbolic Dynamic Programming
  Aswin Nadamuni Raghavan, Alan Fern, Prasad Tadepalli, Roni Khardon, and Saket Joshi (AAAI-12).
  https://www.aaai.org/ocs/index.php/AAAI/AAAI12/paper/download/5012/5336

This simple algorithm is hard to beat for the following reason that connects some details of your egocentric and empathic settings: rather than decomposing a concurrent MDP into independent problems per concurrent action, the optimization of each action (by each advisor) is done in sequence (advisors are ordered) and gets to condition on the previously selected advisor actions.  So it provides an alternate paradigm where advisors actually get to see and condition their policy on what other advisors are doing.  In my own work comparing optimal concurrent solutions to this approach, I have found this approach to be near-optimal and much more efficient to solve since it exploits decomposition.

Why is this relevant to this work?  Because (a) it suggests another variant of the advisor decomposition that at least makes sense in the case of concurrent actions (and perhaps shared actions though this would require some extension) and (b) it suggests there are more options than just the full egocentric and empathic settings in this important class of concurrent action problems that are necessarily solved in practice for large action spaces by some form of decomposition.  This could be an interesting direction for future exploration of the ideas in this work, where there might be additional technical novelty and more space for empirical contributions and observations.

---

### Decision · Program_Chairs · 2018-01-29
**ICLR 2018 Conference Acceptance Decision**

**Decision:**

Reject

**Comment:**

The reviewers agree this is an interesting paper with interesting ideas, but is not ready for publication in its current shape. In particular, there is a need for strong empirical results.